



# Spatiotemporal variability of shortwave radiation introduced by clouds over the Arctic sea ice

Carola Barrientos Velasco, Hartwig Deneke, Hannes Griesche, Patric Seifert, Ronny Engelmann, and Andreas Macke

Leibniz Institute for Tropospheric Research, Leipzig, Germany

**Correspondence:** Carola Barrientos Velasco (barrientos@tropos.de)

**Abstract.** The role of clouds in recent Arctic warming is not fully understood, including their effects on the shortwave radiation and the surface energy budget. To investigate relevant small-scale processes in detail, an intensive field campaign was conducted during early summer in the central Arctic during the Physical feedbacks of Arctic planetary boundary layer, Sea ice, Cloud and AerosoL (PASCAL) drifting ice floe station. During this campaign, the small-scale spatiotemporal variability of global

irradiance was observed for the first time on an ice floe with a dense network of autonomous pyranometers. 15 stations were deployed covering an area of 0.83 km x 1.3 km from June 4-16, 2017. This unique, open-access dataset is described here, and an analysis of the spatiotemporal variability deduced from this dataset is presented for different synoptic conditions. Based on additional observations, 5 typical sky conditions were identified and used to determine the values of the mean and variance of atmospheric global transmittance for these conditions. Overcast conditions were observed 39.6 % of the time predominantly

during the first week, with an overall mean transmittance of 0.47. The second-most frequent conditions corresponded to multi-layer clouds (32.4 %) which prevailed in particular during the second week, with a mean transmittance of 0.43. Broken clouds had a mean transmittance of 0.61 and a frequency of occurrence of 22.1 %. Finally, the least frequent sky conditions were thin clouds and cloudless conditions, which both had a mean transmittance of 0.76, and occurrence frequencies of 3.5 % and 2.4 %, respectively. For overcast conditions, lower global irradiance was observed for stations closer to the ice edge, likely

attributable to the low surface albedo of dark open water, and a resulting reduction of multiple reflections between the surface and cloud base. Using a wavelet-based multi-resolution analysis, power spectra of the time-series of atmospheric transmittance were compared for single-station and spatially averaged observations, and for different sky conditions. It is shown that both the absolute magnitude and the scale-dependence of variability contains characteristic features for the different sky conditions.

# 1 Introduction

The Arctic is a focal point for studying the response of the climate system to anthropogenic forcings (Johannessen et al., 2004). This region is experiencing a rate of warming of surface temperature which exceeds the global average by a factor of



two (Winton, 2006; IPCC, 2013). This leads to thinner (Haas et al., 2008; Lindsay and Zhang, 2005), younger (Maslanik et al., 2007) and less extended (Serreze et al., 2007) sea ice. As the surface temperature increases, snow and sea ice melts, reducing the albedo and increasing the amount of shortwave radiation absorbed by the surface, a process known as the ice-albedo feedback (Curry et al., 1995).

The changes in the shortwave surface radiation budget are inextricably linked to cloud effects (Curry et al., 1996). Kay et al. (2008), using satellite observations from the A-Train, suggested that in a warmer Arctic, shortwave radiation plays an important role in modulating summertime sea ice extent, a factor that can explain the record-breaking 2007 Arctic sea ice extent minimum. Subsequently, Pinker et al. (2014) demonstrated that for the period 2003-2009, the shortwave energy flux into the Arctic ice-ocean system showed similar variations to the sea ice extent loss during the entire melt season and summer
months, with correlations of 0.95 and 0.91, respectively.

    On the other hand, Graversen et al. (2011) concluded that specific areas showing the largest accumulation of shortwave energy did not correspond to negative sea ice concentration anomalies for 2007. Graversen et al. (2008) investigated the link between Arctic warming and changes of atmospheric circulation quantified by the Arctic Oscillation index (AO). They suggested that changes in poleward atmospheric heat transport may be an important cause of Arctic warming.

There is an ongoing debate about the role of external and internal forcings contributing to the amplification of this warming (Kay et al., 2008; Nussbaumer et al., 2012; Perovich et al., 2008; Graversen et al., 2008, 2011), which is often referred to as Arctic amplification. This debate is the motivation to further investigate the influence of shortwave radiation based on observations.

    During summer, shortwave radiation strongly affects the ablation of sea-ice floe edges, and the formation and development
of melt ponds (Light et al., 2008). Clouds contribute to the complexity of the processes by either warming or cooling the surface. Perovich (2018) investigated changes of the surface albedo and net radiative forcing during the Surface Heat Budget of the Arctic Ocean (SHEBA) program (Perovich et al., 1999; Uttal et al., 2002) for sunny and cloudy conditions. He compared 5 pairs of days during summer, concluding that for snow-covered or bare ice conditions, sunny skies are associated with lower radiative heating of the surface than cloudy conditions, oftentimes even showing radiative cooling, due to longwave cooling
dominating over shortwave heating.

    The project **A**rcti**C A**mplification: **C**limate Relevant **A**tmospheric and SurfaCe Processes and Feedback Mechanisms $(AC)^3$ was established to investigate the key processes contributing to Arctic amplification, and to improve our understanding of the major feedback mechanisms, with a particular focus on the influence of clouds. Within $(AC)^3$, the '**P**hysical feedbacks of **A**rctic planetary boundary layer, **S**ea ice, **C**loud and **A**eroso**L**; PS106/1' (*PASCAL*) campaign was conducted. PASCAL took
place from May 24 to June 21 2017 on board of the research vessel Polarstern (Wendisch et al, 2019; Macke and Flores, 2018). The aim of this expedition was to collect observations for the investigation of several processes related to Arctic amplification on a very small scale.

    During PASCAL, an ice floe camp took place from June 4-16, 2017, while *Polarstern* was moored to the drifting ice floe. A multitude of physical, meteorological and biological research observations were conducted. The aim of the present study
is to describe a unique dataset of global horizontal irradiance (GHI) observations obtained from a network of 15 pyranometer



stations, which were operated during the ice floe camp covering an area of about 0.83 km x 1.3 km in order to determine relevant variation under different atmospheric conditions.

This autonomous pyranometer network is a subset of a network of 99 stations which was developed at the Leibniz Institute for Tropospheric Research (TROPOS) to investigate the spatiotemporal variability of global radiation induced by clouds, and

is described in detail in Madhavan et al. (2016). It was first deployed as part of the 'High Definition Clouds and Precipitation for advancing Climate Prediction'$(HD(CP)^2)$ Observational Prototype Experiment (HOPE) field campaigns (Macke et al, 2017). HOPE was conducted to obtain reference observations for evaluating the representation of clouds and precipitation in the German high-resolution atmospheric model ICON-LES (Heinze et al., 2017) within the $HD(CP)^2$ project, in order to improve our understanding of cloud and precipitation processes and their implications for climate prediction. This project

joined the efforts of 19 different institutes in Germany, making possible the synergy between observations and modelling. Within this framework, the pyranometer network was first deployed in the surrounding of Jülich, from April 3 to July 31, 2013, covering an area of 10 km x 12 km (Madhavan et al., 2016; Macke et al, 2017), and again from September 3 until October 14, 2013 around Melpitz, covering a smaller area of 2 km x 2 km (Macke et al, 2017). Two additional field campaigns with the pyranometer network have been subsequently conducted in 2015 and 2018 in Germany.

Analyzing the results from the HOPE-Juelich campaign, Madhavan et al. (2016) concluded that the most significant spatial and temporal variability in the global transmittance was observed during broken cloud conditions. Additionally, a wavelet-based multiresolution analysis was used to obtain wavelet power spectra, which revealed that under this sky condition, significant variability is observed even at frequencies below 1 minute (Madhavan et al., 2017). This conclusion corroborates the findings of Lohmann and Monahan (2018) that, based on the same dataset, suggests to increase the current recording of shortwave

irradiance from one minute resolution as recommmended by the Baseline Surface Radiation Network (BSRN) (McArthur, 2005) to a much higher temporal resolution. Moreover, based on an analysis of the variability of the clear-sky index for HOPE, it was found that cloud-induced shadows do not move fast enough to cover even the shortest of the analyzed sensor pair distances within a second, proposing that to study frequencies of 1s, the pyranometer network would have to be reconfigured to feature much shorter inter-sensor distances to optimally resolve such effects (Lohmann, 2016).

The scope of this paper is to present the dataset obtained from the pyranometer network during the PASCAL ice floe camp, and use it for an analysis of the temporal and spatial variability of the atmospheric global transmittance (ATg) considering synoptic conditions. Section 2 presents the observational data and the methodology. In section 3 the results are presented and case studies under particular sky conditions are analyzed. The paper closes with discussion, conclusions and outlook to future investigations.

## 30  2  Observational data and methodology

The instrumentation and technical specifications of the pyranometer network are described in detail in Madhavan et al. (2016), so only a short summary is given here. In addition, the experimental set up, an update of the calibration, the procedure for quality assurance and the subsequent data processing of the observations are described in the following subsections. The last



subsection also explains the methodology used to classify the ice floe camp period into 5 different sky conditions, which are subsequently used to discuss the pyranometer observations.

## 2.1 Pyranometer network set up

A set of 15 pyranometer stations was installed on an ice floe during the PASCAL ice floe camp (see Fig. 1c). While this

number is relatively small compared to the 99 stations deployed during the HOPE field campaign, the harsh Arctic conditions made their setup and maintenance rather challenging. The area covered had approximate dimensions of 0.83 km x 1.3 km, with individual stations separated by several decameters (see Fig. 1b). The location and spatial distance was limited by other observations which took place on the ice floe, as well as the maximum allowed safe distance from the ship of 1 nautical mile (1.8 km), due to the danger of polar bear attacks. The ice floe drifted from 81.7 to 81.95 degrees northern latitude and from

9.86 to 11.58 degrees eastern longitude during the two weeks of observations (see Fig. 1d).

Each pyranometer station was mounted on an aluminum rod of 1.8 m height (Fig. 1a). On the top, it was equipped with an EKO Instruments silicon photodiode pyranometer with a spectral range of 0.3-1.1 μm (model: ML-020VM), with a data logger and a meteorological station measuring relative humidity (RH) and air temperature (Ta) at 1 Hz frequency. Note that due to the large number of stations, a low-cost pyranometer was used, with an expected accuracy of about 5 %, which is substantially

larger than the accuracy achieved by state-of-the art secondary standard thermopile pyranometers. While ventilation and heating can additionally help to improve data quality in particular regarding dew, rain and frost on the pyranometers, the power requirements for such measures made them unfeasible. It therefore needs to be stressed that the dataset is best suited for the investigation of spatial and temporal variability. Additionally, the data logger contained a Global Positioning System (GPS) antenna module (model: Fastrax UP501) for reliable time and position information.

The accuracy of the Ta measurements are ± 1.5 °C at -40 °C, increases linearly to ± 0.5 °C at 0 °C and remains stable up to 40 °C. Relative humidity observations were flawed due to sublimation occurring on the probe of the sensor. Therefore, the values of this parameter are not considered in this study and are not recommended for use. Finally, power for the system was supplied by a 6 V/19 Ah Zinc carbon VARTA 4R25-2 battery, allowing the system to work continuously for about 7 days. Additional information of the main components of the station are given in Table 1.

### 2.1.1 Update of pyranometer calibration

In 2013, the pyranometers were calibrated with a standard spectrum solar simulator by Eko Instruments after their production (Madhavan et al., 2016). The resulting calibration coefficients are listed in Table 2. The stability of these factors is specified to be better than than 2 % per year. Additional uncertainties are introduced in the measurement system by the data logger and the gain of the instrument amplifier (Madhavan et al., 2016). In order to update the calibration of the sensors, inter-comparison

measurements were conducted in May 2018. The stations were set up in close vicinity to each other for 20 days on the roof of the main building at TROPOS in Leipzig, Germany. Out of this period, 2 hours of observations were chosen on a clear-sky day (10:30-12:30Z). These observations were compared to a recently calibrated secondary standard broadband thermopile pyranometer with a spectral range from 0.3 to 2.8 μm (model: Kipp & Zonen CMP21). After the comparison, the calibration





coefficients were determined to minimize the difference between the network stations and the Kipp & Zonen pyranometer. The updated calibration coefficients are given in Table 2. The mean of the original calibration coefficients was 7.38 $\mu VW^{-1}m^2$, while the updated calibration coefficients had a mean value of 7.19 $\mu VW^{-1}m^2$. This suggests that on average, the sensitivity of the instruments decreased slightly by 2.5 % over time. By updating the calibration, the root mean square error (RMSE)

between the pyranometer stations and the reference pyranometer was reduced from 15.3 $W/m^2$ to 4.5 $W/m^2$, thus showing a significant improvement.

### 2.1.2 Quality assurance

The cleanliness of the sensor's glass dome, and the horizontal alignment of the sensor leveling plate are important factors to guarantee the accuracy of the global horizontal irradiance (GHI) observations. In order to assure the quality of the dataset,

the stations were checked daily recording the status of the leveling and cleanliness. Figure 2 shows the general status of the stations. Every station is shown on the Y-axis, while the date is given on the X-axis. Each square is divided in two triangles, the upper one representing the leveling status of the station, and the lower one the cleanliness status. The green color shows a well-leveled and clean station, yellow is used for a partially leveled station or a dome with liquid droplets, and red is used for completely unleveled stations and an iced dome.

Complete misalignment was found to occur when the base of rod was no longer properly supported by the snow due to melting, leading to a tilt of the sensor and systematic errors in particular for direct sunlight. Icing of the radiation sensor dome occurred when moist air masses with supercooled water droplets were present, which froze upon impact with the dome (Van den Broeke et al., 2004). This condition will generally lead to an underestimation of GHI. Our following results do not make use of observations with iced domes or completely unleveled stations (red flags).

It is nevertheless possible that the domes were contaminated by droplets or ice before or after the daily quality assurance checks. However when significant fluctuations of atmospheric global transmittance (ATg) occurred, these were verified with observation by the all-sky camera operated on *Polarstern*. Such moments are discussed in the following sections.

### 2.1.3 Data processing

The raw data were stored by the stations in ASCII files in counts of 10 bits on an SD card. The records were subsequently

converted to GHI ($W/m^2$), air temperature (Ta in K) and relative humidity (RH in %) using the equations given in Madhavan et al. (2016). The values of GHI, Ta and RH were averaged to 1Hz sampling frequency according to the GPS time reference. The dataset was processed into NetCDF files following the latest 1.7 version of the Climate and Forecast Conventions (Unidata, 2012). Daily files containing all 15 pyranometer stations including GHI, Ta, RH, latitude (in degrees), longitude (in degrees), leveling and cleanliness flags, station number, corrected calibration coefficient were prepared. Earth Sun distance (in AU), solar

constant and solar zenith ($\theta$ in degrees) and azimuth angles ($\alpha$ in degrees) were added to the datad-set based on the algorithms given in the World Meteorological Organization (WMO) Guide to Meteorological Instruments and Methods of Observations for practical application (WMO, 2008).





For our analyses, ATg was calculated from the GHI measurements following equation 1, where $\epsilon$ is the actual Earth-Sun distance (in AU), $F_o$ is the solar constant, with a value of 1360.8 $W/m^2$ (Kopp and Lean, 2011) and $\mu_o$ is the cosine of the solar zenith angle ($\theta$ in degrees). ATg is hence defined as the fraction of radiation that is transmitted through the atmosphere (e.g. Liou, 2002).

$$ATg = \left[\frac{F}{\epsilon^2 \cdot F_o \cdot \mu_o}\right] \tag{1}$$

At the top of atmosphere (TOA) ATg is 1, whereas at the bottom of the atmosphere (BOA) the value is reduced due to absorption and scattering within the atmosphere. At the BOA the highest values are generally observed for cloudless conditions, while clouds generally cause a reduction of ATg. Under certain situation however, the presence of broken clouds can amplify ATg to reach values larger than 1 due to multiple scattering. Such enhancements can excceed 400 $W/m^2$ and persist up to 20 seconds

(Schade et al, 2007).

### 2.1.4 Sky classification

To analyze the effect of clouds on the ATg, a general overview of cloud conditions was obtained first. An overview of the cloud conditions present during PASCAL was already given by Wendisch et al (2019). Whereas their classification was based on time-height cross sections of clouds classified from vertical-stare active and passive remote sensing observations, the classification

used here is based on the visual inspection of all-sky camera observations recorded aboard Polarstern. These images provide a more complete impression of the horizontal variability of clouds which is important for the characterization of inhomogeneities of the atmospheric transmittance. Initially, a separation of overcast, cloudless and broken cloud conditions was made from the all-sky camera observations on *Polarstern*. Due to the availability of observations with a cloud radar of type Mira-35, and multi-wavelength polarization Raman lidar Polly-XT (Engelmann et al., 2016) during PASCAL (Wendisch et al, 2019; Macke

and Flores, 2018), these were used in addition to identify the presence of multi-layer clouds, a separation which has not been considered in previous studies (Madhavan et al., 2016, 2017; Lohmann, 2016; Lohmann and Monahan, 2018). Cirrus clouds and low, geometrically thin stratus clouds were also observed, and have been considered together as thin clouds due to their low occurence frequency to study their effects on ATg.

The final classification was made considering the daily quicklooks of the range-corrected signal at 532 and 1064 nm wave-

length from the lidar, daily plots of the radar reflectivity, and images from the all-sky camera. Whenever the lidar signal was attenuated, the information of the cloud radar was used to identify the presence of single or multiple layers of clouds. For apparently clear-sky moments as identified by the lidar and radar quicklooks, images from the sky camera were used to confirm the absence of clouds, or to identify thin cloud layers or broken cloud situations. In this study, conditions were only considered to be cloudless when the images from the all-sky camera did not show any clouds within its fisheye field-of-view. The transition

from one type of sky condition to another was only recorded when this class lasted for longer than 15 minutes. Cases, when the change was observed for less than 15 minutes were assumed to belong to the previous sky condition. Thin clouds were defined as clouds with a height difference between the cloud base and top of below 450 m. The result of this classification is shown by the color coding in Figure 4, and examples for each condition are shown by all-sky camera images for each case study.



## 3 Results

In this section, results obtained from an analysis of the pyranometer network observations are presented. First, the synoptic conditions during the ice floe camp are described, and related to the observations of atmospheric transmittance under specific sky conditions. Case studies pertaining to each of the used sky conditions are presented next. Finally, the difference in power
spectra for these sky conditions obtained from a wavelet-based multiresolution analysis of atmospheric transmittance are discussed.

### 3.1 Synoptic and near-surface temperature classification during PASCAL: Ice floe camp

A detailed synoptic scale description for PASCAL is provided in Knudsen et al. (2018) where a longer period covering the airborne activities of the **A**rctic **CL**oud **O**bservations **U**sing airborne measurements during polar **D**ay (ACLOUD) campaign is
analyzed. Three key periods were characterized based on near-surface and upper-air meteorological observations, operational satellite and model data. The PASCAL observations at the ice floe camp took place mainly in the warm period, May 30 to June 12, when warm and moist maritime air intruding from the south and east dominated the synoptic conditions. However, the classification by Knudsen et al. (2018) considers a larger spatial scale that neglects sensitive local episodes at the ice-floe camp which need to be considered when analyzing the near-surface temperature changes at the ice floe camp. Therefore, the
classification has been sharpened for the synoptic conditions during the limited period of the ice floe camp.

With the aim to consider the influence of the large-scale Arctic circulation on the synoptic conditions during the ice floe camp, the Arctic oscillation index (AO) is used in our analysis here. It refers to an opposing pattern of pressure difference between the Arctic and the northern mid-latitudes.

The AO is a climate index which characterizes the atmospheric circulation of the Arctic region by considering geopotential
height anomalies of the 1000 mb isobar between $20° − 90°N$. It is defined as the loading of the dominating mode obtained from an Empirical Orthogonal Function (EOF) analysis of the monthly-mean anomaly fields. Here, daily values of the AO index as reported by the Climate Data Center of the National Oceanic and Atmospheric Administration are used (source: ftp://ftp.cpc.ncep.noaa.gov/cwlinks/). To determine the loadings, the current height anomaly from the monthly average field is projected onto the pattern of the leading EOF mode, which has been determined from monthly averages obtained from the
NCEP/NCAR reanalysis for the period 1970-2000.

A positive AO index generally corresponds to strong westerly winds in the upper atmosphere, lower than usual atmospheric pressures and temperatures in the Arctic, and an opposite effect on pressure and temperature mid-latitudes. On the other hand, a negative AO index leads to weaker upper-level winds, higher atmospheric pressure and warmer temperatures in the Arctic, and contrary effects and an increase of storms in the mid-latitudes (Thomson and Wallace, 1998).
A classification is presented considering the near-surface air temperature and contrasting it to the AO index (Fig.3a and 3b). One can distinguish two main periods. First, from June 4-9, 2017, a cold period was observed with relatively low near-surface air temperatures (mean 269.7 K). The negative AO index suggests a reduction in the difference of surface pressure between the Arctic and northern middle latitudes, resulting in a larger polar low pressure system and warmer-than-usual temperatures





(Talley et al., 2011). From June 10-16, 2017, a warm period can be identified due to an increase in temperature (mean 272.32 K). The temperature reaches two maxima on June 10 and 13. A positive AO index is generally associated with colder-than-usual temperatures, stronger westerly winds in the upper atmosphere, and a more cyclonic circulation that can influence sea ice motion (Rigor et al., 2002). Even though the near-surface temperature shows the two periods mentioned, the AO index

suggests that the cold period occurred in conditions warmer than average and the warm period in conditions colder than the average over the Arctic for the days observed.

Figure 3c shows that humid conditions prevailed during most of the ice floe camp, with a notable drop in relative humidity during the transition between the cold and the warm period. Two relative humidity (RH) sensors were operated at different heights on *Polarstern*. One sensor was mounted on top of the OCEANET facility OCEANET at about 10 m above sea level,

while the second one was installed on the crow nest of *Polarstern* at a height of 29 m. The discrepancy in the observation of both sensors are mainly due to the different heights and locations of the sensor. The *Polarstern* sensor, a Vaisala sensor of type HMP155 (accuracy: < ±5.0% RH), was exposed to more open conditions, whereas the OCEANET sensor of type EE33 (accuracy: ±1.3% RH), was relatively more protected (explanation by Henry Kleta. Deutscher Wetterdienst - DWD). Considering the RH values obtained by the *Polarstern* sensor, the mean RH during the first period was 94 %, and 95.7 % for

the second. On June 10, 2017, the longest cloudless period occurred, and the RH observed by the *Polarstern* sensor dropped to 80%. After this event, a moist air intrusion was observed already in the evening of June 10th (Knudsen et al., 2018).

About 97.2 % of the time, cloudy skies were observed during the ice floe camp. Low-level clouds with cloud tops no higher than 2.6 km and a mean cloud base of 220 m above the surface were observed. During the cold period, mostly overcast conditions with single layer clouds were present, whereas multi-layer and broken clouds dominated the second period (Fig. 4).

During the first and the second period, mean wind speeds of 4.77 and 5.32 m/s, respectively, were observed (Fig. 3e). Four maxima in wind speed were observed during the ice floe camp, on June 4 with southerly winds, on June 7 with northerly and easterly winds, on June 11 with southerly winds, and on June 14 with northerly winds (Fig. 3d and e).

## 3.2   Meteorological classification of global transmittance

Using the methodology introduced in Section 2.1.4, Figure 4a shows the time series of the ATg for the whole period of the ice

floe, and considering all operational pyranometers. Figure 4b shows the time series of the inter-station standard deviation (SD), and Figure 5 presents the histograms for each of the sky conditions.

As can be seen in Figure 4, about 40 % of the time, overcast conditions dominated the ice floe camp, occurring mainly during the cold period. This sky condition resulted in ATg values generally lower than 0.7, with a mean value of 0.46 (Fig. 5 and Table 3). The monomodal distribution of ATg was mainly characterized by stratus clouds, however, during June 9 and 11,

the presence of stratus nebulosus clouds was observed due to the formation of rain or drizzle as visible from the all-sky camera.

Multi-layer clouds, mainly consisting of double layers (Fig. 3d, Fig. 4a), occurred about 30 % of the time. They again caused a monomodal distribution of ATg, with a mean value of 0.43 (Fig. 5 and Table 3). This sky condition was more frequently observed during the warm period. The time-height cross section of the radar reflectivity revealed complex structures of three layers of clouds on June 13 and 14, and two layers reaching up to 9 km height observed on June 15. On June 9 from 9:50Z to





12:00Z, and from June 12 at 19:20Z to June 13 at 9:15Z, precipitation was observed by the all-sky camera. During this episode, it is possible that the domes of the pyranometer were covered with small droplets which might not have been present at the moment of daily quality control.

The third most frequent sky condition corresponded to broken clouds (22.1%). This sky condition was present during both
the cold and the warm period, and it showed subtle differences that complicated its classification. June 7 had less frequent cloud gaps than other days. The sky was covered by stratus fractus clouds as seen in Figure 13a at 12:06Z. June 8 showed a combination of cumulus fractus (Fig. 13b at 10:14Z) and stratus fractus clouds. At the end of June 10, broken cloud conditions were observed, corresponding also mainly to stratus fractus clouds. On June 11 and 12, sky conditions were similar to June 7, with slight precipitation from June 11 at 21:15Z to June 12 at 1:10Z. The ATg observed for broken clouds had a monomodal
distribution and a relatively high mean ATg of 0.61, with a SD of 0.146 (Fig. 4, Fig. 5 and Table 3).

The selection of thin cloud conditions was rather complex, due to their similarities with broken clouds, and because they frequently occurred outside the field of view of the lidar. For these cases, the classification was based entirely on the all-sky camera. On June 9, a fairly uniform stratus cloud was observed from 16:35 to 18:08Z, and from 23:24Z to June 10 at 7:18Z, when cirrus fibratus clouds at 10 km height occurred. Thin-cloud conditions appeared only during the transition period, with
a frequency of only 3.5% (Fig. 3), and a mean ATg of 0.76 (SD 0.043). Finally the least-frequently occurring condition with only 2.4% of occurrence frequency was cloudless sky, with a monomodal distribution and a mean ATg of 0.76 (SD of 0.028). This sky condition only occurred at the end of June 9, and on June 10 from 11:10Z to 15:43Z.

The following sub-sections analyze particular case studies for the different sky conditions discussed. The analysis takes into account the temporal standard deviation of ATg of the the stations with reliable observations, the histogram of the distribution
of global transmittance, a map for comparing the spatial patterns of the standard deviation, a wind rose to indicate the wind speed and direction, and a photograph of the all-sky camera to visualize the corresponding cloud structure.

### 3.3   Cloudless case - June 10, 2017

The day with the longest cloudless period was June 10, 2017 (11:10Z - 15:43Z). This episode also coincides with the beginning of the warm period (Section 2.1), with a mean ambient temperature of 274 K. This day was also characterized by the warmest
air in the upper atmospheric layers during the ice floe period, and a relatively low boundary layer of about 200 m height (Knudsen et al., 2018).

The positive AO index (Fig. 3a) suggests dry conditions (Fig. 3c) associated with an anti-cyclonic circulation (Knudsen et al., 2018), leading to reduced cloudiness and enhanced downward shortwave radiation (Kay et al., 2008). Figure 6a shows the time series of the global transmittance, and the green background highlights the cloudless period. In Figure 6b, the narrow
distribution of ATg is shown, with values just below 0.8. The comparatively small reduction of transmittance is mainly due to scattering and absorption by atmospheric gases, and, to a lesser degree, due to aerosols. In early spring or autumn, lower values of ATg are expected due to the lower sun and the resulting longer optical path through the atmosphere (Zhao and Garrett, 2015). The temporal standard deviation of the operational pyranometers is shown in Figure 6d, having a mean value of 0.014 for the period of interest. Figure 6d shows the standard deviation of the measurements for each pyranometer during the selected





period. Variability in the observed global transmittance is mainly noticed for stations 39 and 42, with a standard deviation of 0.0090 and 0.0092, respectively. It is likely that this variability can be attributed to undetected deficiencies such as an unleveled instrument due to melting.

### 3.4 Overcast case - June 6, 2017

The overcast case selected occurred on June 6 between 00:00Z and 23:59Z. During this day, the boundary layer increased in height from 300 m to about 430 m. This day was characterized by a mean cloud base and top of about 100 m and 490 m, respectively (Wendisch et al, 2019). The wind speed measured on-board *Polarstern* increased from 2.5 m/s to almost 10 m/s in the evening with easterly origin (see Fig. 3e and 7f). The mean surface temperature recorded by the pyranometer network during this period was 270.1 K, with an inter-station SD of 0.38 K.

The ATg during this period showed a monomodal distribution (Fig. 7b), with a mean value of 0.50. The time series of the standard deviation presented values lower than 0.1 (mean SD = 0.023), as well as high values from 7:29Z to 8:27Z, when the dome of the instruments might have experienced momentary icing (Fig. 7d). The stations showing this behaviour were 26, 30, 34, 35, 38, 40 and 43. The spatial distribution of the SD shows slightly higher values in the south-west of the ice floe region (Fig. 7e). This small variance is most likely attributed to a more snowy surrounded area, whereas pyranometers 37, 34, 40, 39, 42, 43 and 44 were closer to the ice edge (Fig.1b). Snow-covered surfaces reflect more SW radiation to the atmosphere than darker surfaces such as the open ocean. Once this reflected energy reaches a cloud, a part of it is reflected back to the surface increasing the total amount of downward SW radiation. Thus, for homogeneous single-layer clouds, the stations closer to the ice edge show lower variation because more energy is absorbed by the darker surface.

In addition, Figure 8 shows the spatial distribution of the mean ATg. It clearly shows how stations near *Polarstern* (red star) and the edge of the ice floe observe lower values of transmittance than the stations farther away from the ice edge, with absolute differences ranging up to 0.06. A likely explanation is the low surface albedo of the open water, which leads to a reduction in multiple reflections between the ground and cloud base compared to those stations fully surrounded by snow-covered ground. A similar plot was made for June 5, where the sky conditions were also overcast. A similar, but a less pronounced pattern was observed (not shown), with smaller difference of up to 0.03.

### 3.5 Thin clouds case - June 9, 2017

Thin clouds were not frequent during the ice floe camp period. The only cases occurred on June 9 and 10 (Wendisch et al, 2019). The thin-cloud case selected for the case study was observed between 16:36Z and 18:08Z on June 9. The overall cloud conditions during this period were complex, because the clouds varied in both height and depth (See Fig. 3). From 16:30Z to 17:00Z, a single cloud layer with a base height of 1.1 km and top height of 1.5 km was observed. During this period, drizzle was observed at the cloud base. Following this period, a very shallow cloud was observed with a cloud top of about 0.45 km from 17:00 to 18:05Z.

The mean ambient temperature during this period was 273.9 K with a SD of 0.35 K. During this condition the cloud-base height was at 450m, the temperature increased steadily from 273.15 to 274.11 K. The winds came mainly from the south,



following an anticyclonic circulation with mean wind speeds of 5.2 m/s (Knudsen et al. (2018), Fig 3e, Fig. 9f). The variance of height and composition had an impact on the ATg and temperature measured by the pyranometer network as can be seen in the shaded purple background in Figure 9a. The highest values of inter-station SD occurred at the moment when the optical thickness of the cloud decreased at 17:30Z. The mean ATg during this period was 0.72, with a left-skewed distribution (see Fig.

9b). The spatial distribution of the SD shows a similar trend as in the overcast case. This suggests that during these conditions, the highest variability occured in the region away from the ice floe edge and melt pond (Fig.1b). The spatial distribution of mean transmittance shows a less evident pattern, suggesting that part of this variability is caused by variable cloud structures more than due to the particular location of the stations. The highest difference of the mean transmittance between all operational stations is 0.08.

## 3.6 Multi-layer case - June 13, 2017

The multi-layer case selected was June 13, 2017, from 00:00Z to 19:15Z. This day had a positive AO index, implying that the episode was dominated with by an anticyclonic circulation, when air was advected over the open ocean, favoring high temperature and humidity (Knudsen et al. (2018), Fig. 3a). Wind speeds were below 5 m/s coming from the south-west and north-east (see Fig. 3e and Fig. 10f). The RH was rather unstable, varying from 85 % to 96% at 10 m height and close to 100

% at 29 m high (Fig. 3c).

June 13 was the warmest day of the ice floe camp. At around 7:30Z and 10:15Z, the highest temperatures of 279.03 K and 281.1 K were recorded (See Fig. 11). This increase in temperature was not recorded by the *Polarstern* temperature sensor (see Fig. 3b). The observed fluctuations suggest that the near-surface air temperature over the ice floe experienced significant variability, likely due to turbulent mixing of with warmer, elevated air masses. A spatial comparison among the sensors of

SD and mean near-surface temperature did not indicate a particular pattern, but showed relatively high differences of mean temperature of up to 1.4 K.

From 00:00Z to 02:24Z, three layers of clouds were identified with tops at 0.9, 4.8 and 7.8 km. The remaining time of the period was characterized mainly by complex structures and two cloud layers showed low values and a right-skewed distribution, having a mean of 0.45 (SD 0.0165) (Fig. 10b, Table 3). The time series of ATg and inter-station SD indicated no significant

variation among the stations (Fig. 10a and d). Furthermore, the values of temporal standard deviation found for the individual stations do not show a prominent spatial variation as, for example, observed for the overcast and thin cloud cases (Fig. 10e). Considering that the difference between the lowest and the highest value of SD is low (0.007), the variation of the global horizontal irradiance for this case is negligible.

### 3.7 Broken clouds case - June 8, 2017

The period from 8:30Z to 18:59Z of June 8, 2017, was characterized by fluctuating occurrences of stratus fractus (Fig. 13b at 11:02Z) and cumulus fractus (Fig. 12c). During this case south-easterly winds with a mean speed of 2.6 m/s prevailed (Fig. 3c and Fig. 12f). A remarkable drop of relative humidity was observed by the OCEANET sensor at 10 m height at around 12:00Z, whereas the values at 29 m height remained stable (Fig. 3c). Near-surface air temperatures observed with the pyranometer




network increased by 2 K from 267 to 269 K. This behavior was also recorded by the *Polarstern* sensor (Fig. 3b). Before the broken cloud conditions began, the temperatures recorded were the lowest registered during the entire campaign.

The pink-shaded background in Figure 12a and c highlights the broken cloud period. The mean and median ATg is represented in Figure 12a in blue and red, respectively. The gray spikes represent the minimum and maximum values recorded for

each time, indicating that for instances the transmittance varied from 0.4 up to and surpassing 1 for a few stations. The increase of diffuse shortwave radiation is connected to the heterogeneity of these cloud fields and is due to cloud scattering. Under these conditions, the plane-parallel cloud approximation cannot satisfactorily describe radiative transfer (Wendler et al., 2004; Schade et al, 2007). In particular, it is well-known that horizontal photon transport can lead to periods with enhanced solar radiation, where values of the global transmittance can exceed the clear-sky values or even unity for some moments (Schade et

al, 2007). In addition, Byrne et al. (1996) demonstrated that in broken clouds fields, the average photon path-length is greater than that predicted by homogeneous radiative transfer calculations, also leading to an enhanced absorption.

Based on the mean values of ATg for this period, the histogram in Figure 12b shows a left-skewed distribution and a mean value of 0.81, with a mean standard deviation of 0.01. Fluctuations of the standard deviation can be easily recognized in Figure 12d, with more noticeable spikes before and after the selected period. The spatial variability, shown in Figure 12e, indicates

higher values of temporal standard deviation for the stations further away from the ice floe edge, however, the mean ATg does not show the same pattern as in Figure 8. The latter indicates that the variability observed is dominated by the cloud organization and not by the contrast between the open ocean and highly reflective surfaces.

## 3.8 Wavelet-based multiresolution analysis

To investigate the time-scale dependence of variability in global irradiance, the time series of ATg obtained from the pyranome-

ter network have been subjected to a wavelet-based multiresolution analysis using the Haar wavelet, following the methodology introduced in Deneke et al. (2009) and Madhavan et al. (2017). In summary, $J = 13$ lowpass-filtered versions of the time-series are calculated first, using a running mean of length $L = 2^J$ as a filter. In wavelet analysis, these running means are referred to as the wavelet smooths. The difference between two wavelet smooths of scale $J$ and $J + 1$ correspond to the result of a bandpass filter and are called wavelet details. The wavelet details are then used to obtain time-localized estimates of the time-

scale-dependent variance of the time series (Percival, 1995), which is denoted as the wavelet power spectral density (WSD).

Figure 14 shows the WSD obtained from the observations for a period of three hours of broken clouds, multi-layer clouds, and overcast conditions from 15:00Z to 18:00Z for the case days presented previously, in a double-logarithmic plot, and includes estimates of its uncertainty. Figure 14 compares the WSD calculated in two different ways. Filled circles correspond to the WSD which has been calculated based on the average ATg of all functional stations, which approximates the WSD of ATg

averaged across the spatial domain of the network, with a characteristic length scale of about 1km. Empty circles correspond to the averaged WSD of individual stations and thus a point-like measurement. Due to the 3-hour length of the time series, the WSD for periods at and above 1000 s cannot reliably be reliably observed, as is evident from the increasing uncertainty.

A characteristic decrease of variance with increasing temporal frequency (or equivalently, a decreasing time period) is observed for the different sky conditions. In particular, strongly differing slopes of the WSD are observed for the different





conditions and frequency ranges, which suggests that the WSD is sensitive to structural differences of the clouds. As noted already by Madhavan et al. (2017), variability is significantly reduced when considering the spatially averaged atmospheric transmittance, irrespective of the considered cloud type. Broken clouds exhibit the largest variability, while the multi-layer cloud situations show the smallest variability across all time scales.

In Figure 15, the WSD for two different periods classified as broken clouds are compared. As noted before, on June 7, stratus fractus were observed, with stronger winds likely responsible for stronger fluctuations and longer periods of cloud-free sky, while the broken clouds observed on June 8, corresponding to a mixture of cumulus fractus, stratus fractus and stratocumulus, introduced less fluctuations in ATg. The lower variability can already be seen in Figure 4b. It is noteworthy that spatial averaging has a much stronger effect on the magnitude of variability for the June 8 case, which indicates that the relevant variations in cloud properties occurred on length scales smaller than the extent of the pyranometer network, while a much smaller reduction is observed for June 7. This indicates that cloud scales larger than the extent of the pyranometer network dominated the variability in transmittance during this period.

As noted before by Madhavan et al. (2017), a stronger scale-dependency can be recognized in the WSD for the broken clouds observed on June 8 compared those of June 7. Estimating the slope of the WSD from the 4 points above and below a period of 100s, slopes of the WSD of 1.5 and 1.7 are obtained for June 7, while much lower values of 0.9 and 1.0 are found for June 8, for spatially averaged and point observations, respectively. Here, the values for June 7 are close to the theoretical value of $5/3$ expected from turbulence theory for the dissipation of energy expected from homogeneous isotropic turbulence, while the lower values are in better agreement with the values reported by Madhavan et al. (2017) for broken cloud observations.

## 4 Discussion, conclusion and outlook

Over the past years, the Arctic has been experiencing an unprecedented increase in surface temperature and an associated decrease in sea ice extent, exceeding model-based climate projections by far. Scientific efforts to identify and understand the mechanisms that contribute most to this Arctic warming are still ongoing. After the two extreme events with very low sea ice in 2007 and 2012, the debate to explain these events was principally divided into two sides. Several studies suggest that meridional heat transport is a main contributor to the Arctic warming (Nussbaumer et al., 2012; Graversen et al., 2011). On the other hand, several studies propose that anomalies in the shortwave radiation budget and clouds contribute to sea ice loss during summer (Kay et al., 2008; Pinker et al., 2014).

As one specifc aspect of shortwave radiation budget, the present study focuses on the analysis of the spatiotemporal variability of the shortwave irradiance at the surface as it is introduced by clouds. To support our analysis, the characterization of synoptic conditions given by Knudsen et al. (2018) is used as basis. Focusing on the near surface air temperature, we identified a cold period from June 4-9, 2017, and a warm period from June 10-16, 2017. Although the classification labels the first period as cold, the AO index indicates that atmospheric conditions were warmer than average. In contrast, atmospheric circulation over the Arctic for the warm period featured stronger westerlies at subpolar latitudes and lower sea level pressure over the Arctic (Thomson and Wallace, 1998; Rigor et al., 2002).



During the cold period, overcast conditions with single layer clouds prevailed, with air masses coming mainly from the East and South. The mean ATg during overcast conditions was relatively low (0.48). The warm period was mainly dominated by multi-layer clouds, with a mean ATg of 0.41, and with winds mainly coming from west and north. The distribution and temporal variability of ATg for overcast and multi-layer clouds were found to be similar. However, the distinction is important

since overcast conditions represent more clearly the diurnal cycle and the spatial distribution of the stations showing a specific pattern. Broken clouds were observed during both periods and for wind speeds higher than 4 m/s, and air masses coming from the north and east for the cold period, and from the west during the warm period. The mean ATg for broken clouds observed was 0.61, and showed the highest temporal variability.

Wavelet-based power spectra showed pronounced differences for different sky conditions, with highest variances for low

broken clouds. Considering two different periods of broken clouds, different scaling properties were observed, likely reflecting different typical scales of cloud structures. The variances observed during broken cloud conditions however seem to be smaller than the ones reported during a field campaign in Germany in 2013 by Madhavan et al. (2017), likely due to less convective cloud development taking place in the Arctic. Additionally, we studied the differences between broken cloud conditions. The mixture of stratus fractus, cumulus fractus, and stratiformis clouds marked a higher variability computed with the WSD than

just stratus fractus cases that were more recurrent during the ice floe camp. This difference is relevant to better characterize the variability of shortwave radiation and link it to cloud structure for example in radiative closure studies considering 3D radiative effects (Rozwadowska and Cahalan, 2002).

For single layer clouds the location of stations showed a pattern making a division between the stations near the ship (ice floe-edge) and the ones located farther. This behavior suggests that highly reflective surfaces enhance the spatial variability for

the cases studied. It should be noted that this pattern is based on the limited dataset of 2 weeks and a more solid conclusion should be made after analyzing a larger period of time.

Although a comparison of ATg including its spatio-temporal variability was made for different sky conditions, its relevance for the shortwave radiation budget and the Arctic climate system cannot be assessed based on the present, rather short time series of observations. Specific and relevant analysis of the snow metamorphism is still needed in order to understand thermal

diffusivity and consequently the effect on ice thickness evolution (Saloranta, 2000). Transmission and absorption of shortwave radiation by Arctic surface are equally important, not only to study sea ice sea alteration (Light et al., 2008; Nicolaus et al., 2012), but also to better understand the direct influence on bio-geochemical processes that depend on sub-ice light conditions (Slagstad et al., 2011).

The pyranometer network offers valuable information on the variability of the shortwave irradiance at the surface on small

scales, making it possible to better characterize temporal and spatial fluctuations than from single station measurements. In the future, we plan to use this dataset as a reference for comparison with radiative tranfer simulations using a 3-D Monte Carlo radiative transfer model using large eddy simulations as input, which are also being conducted within the scope of the $(AC)^3$ project. This will allow to investigate and better understand the link between cloud spatial structure and the resulting variability in the shortwave radiation field. Future work also aims to investigate radiative closure studies and cloud radiative effects based

on the ground-base remote sensing observations conducted aboard *Polarstern* using a 1-D radiative model.





*Data availability.* The data published is available on https://doi.pangaea.de/10.1594/PANGAEA.896710

*Author contributions.* Carola Barrientos Velasco led the design, coordination, writing process of the manuscript, collected and analyzed the data. Hartwig Deneke contributed with the analysis of the study, design, advised during the writing process and provided a descriptive text for the multi-resolution analysis. Patric Seifert and Hannes Griesche evaluated the manuscript, and along with Ronny Engelmann provided
5  advice on the use of ancillary observations for the sky classification. Andreas Macke along the other authors contributed to the subsequent improvement of the analysis and the manuscript.

*Competing interests.* The authors declare that they have no conflict of interest.

*Disclaimer.* TEXT

*Acknowledgements.* We gratefully acknowledge the funding by the Deutsche Forschungsgemeinschaft (DFG, German Research Foundation)
10  – Project Number 268020496 – TRR 172, within the Transregional Collaborative Research Center "ArctiC Amplification: Climate Relevant Atmospheric and SurfaCe Processes, and Feedback Mechanisms $(AC)^3$". We thank Captain Thomas Wunderlich and the entire crew of *Polarstern* for their logistic support. We also thank our colleagues at AWI, DWD, Leipziger Institut für Meteorologie and TROPOS for their logistic support and scientific cooperation.



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



**Table 1.** Main components and specifications of a pyranometer station

| 1. Photodiode pyranometer sensor | Characteristics (ML-0.20VM) |
|---|---|
| Response time | 10ms |
| Zero offset - thermal radiation (200 $W/m^2$) | 0 $W/m^2$ |
| Zero offset - temperature change (5 K/h) | 0 $W/m^2$ |
| Non-stability [a] | ± 2 % |
| Nonlinearity [b] | < 0.2% |
| Temperature response [c] | ± 0.5 % |
| Spectral error (during the day) | ± 2-5 % |
| 2.ADC data logger | Characteristics (Driesen+Kern DKRF 4001-P) |
| Analog power supply output | 3.3 V |
| Temperature range | -40 to 85°C |
| Differential linearity error (resolution) | ± 1 Least significant bit (LSB) |
| Gain error | ± 5 % |
| 3. Amplifier | Characteristics (INA 333) |
| Operational temperature range | -40 to + 150 °C |
| Power supply voltage range | 1.8 - 5.5 V |
| Range of gain | 1 to 1000 |
| Gain error | ± 0.3 % (Gain = 300) |
| 4. GPS | Characteristics (FasTrax UP501) |
| Position accuracy | 1.8 m (CEP95) |
| Velocity accuracy | 0.1m/s |
| Time accuracy | ± 50 ns(RMS) |

[a] % change in responsivity per year)

[b] % deviation from responsivity at 1000 $W/m^2$ due to change in irradiance %

[c] % deviation due to change in ambient temperature from -10 to 50 °C





**Table 2.** Calibration coefficients ($K_c$) [$\mu V W^{-1} m^2$] and root mean square error (RMSE)[$W/m^2$] before (1) and after (2) the correction for each station number (Stn)

| Stn | 24 | 25 | 26 | 30 | 32 | 33 | 34 | 35 | 37 | 38 | 39 | 40 | 42 | 43 | 44 | Mean |
|---|---|---|---|---|---|---|---|---|---|---|---|---|---|---|---|---|
| $K_{c1}$ | 7.55 | 7.22 | 7.35 | 7.22 | 7.49 | 7.44 | 7 | 7.35 | 7.54 | 7.44 | 7.43 | 7.32 | 7.35 | 7.15 | 7.09 | 7.38 |
| RMSE$_1$ | 29.0 | 15.6 | 5.6 | 11.2 | 12.3 | 6.5 | 19.9 | 24.0 | 17.2 | 14.7 | 13.5 | 15,8 | 20.3 | 1.7 | 19.8 | 15.13 |
| Bias$_1$ | 14.5 | 7.5 | 2.6 | 5.5 | 6.0 | 2.5 | 9.9 | 11.9 | 8.5 | 7.2 | 6.6 | 7.6 | 10.1 | 0.2 | 9.8 | 7.3 |
| $K_{c2}$ | 7.26 | 7.06 | 7.37 | 7.30 | 7.35 | 7.36 | 6.80 | 7.10 | 7.36 | 7.30 | 7.27 | 7.15 | 7.15 | 7.15 | 6.9 | 7.19 |
| RMSE$_2$ | 4.7 | 5.8 | 3.5 | 2.8 | 5.0 | 5.8 | 5.1 | 6.2 | 4.0 | 2.2 | 5.9 | 6.6 | 3.9 | 1.7 | 4.5 | 4.5 |
| Bias$_2$ | 2.00 | 2.03 | 1.40 | 0.77 | 2.04 | 2.08 | 2.35 | 2.71 | 1.75 | 0.83 | 2.64 | 2.42 | 1.3 | 0.16 | 1.72 | 1.76 |



**Table 3.** Mean values of: Ambient temperature [Ta], atmospheric global transmittance (ATg) [-], temporal standard deviation of transmittance SD(ATg) [-] for case studies (C) and all period (P). All based on the pyranometer network.

| | Overcast | | Broken clouds | | Thin clouds | | Cloudless | | Multi-layer | |
|---|---|---|---|---|---|---|---|---|---|---|
| Case/Period | C | P | C | P | C | P | C | P | C | P |
| Date | June 6 | - | June 8 | - | June 9 | - | June 10 | - | June 13 | - |
| Time [UTC] | 00:00-23:59 | - | 8:55-19:00 | - | 16:36-18:08 | - | 11:10-15:43 | - | 00:00-19:15 | - |
| Ta [K] | 270.2 | 271.1 | 269.0 | 271.1 | 273.9 | 274.2 | 274.3 | 274.3 | 274.6 | 271.93 |
| ATg [-] | 0.50 | 0.46 | 0.77 | 0.61 | 0.72 | 0.76 | 0.78 | 0.76 | 0.46 | 0.43 |
| SD(ATg) [-] | 0.071 | 0.084 | 0.118 | 0.146 | 0.080 | 0.043 | 0.015 | 0.028 | 0.107 | 0.114 |

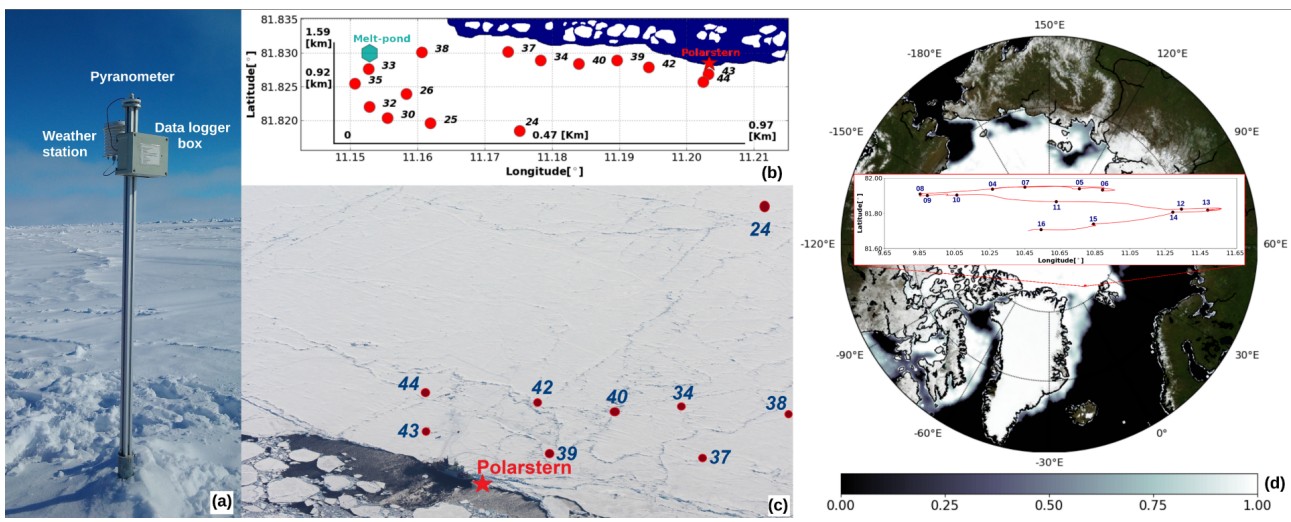

**Figure 1.** (a) Photograph of a pyranometer station on the ice floe. (b) Map of the pyranometer stations. Red circles show the location on June 11, 2017, at 14:50Z, while the red star marks the position of *Polarstern*, and the turquoise hexagon marks the approximate position of a melt-pond. (c) Edited photograph of the ice floe station showing the approximate location of several stations (red circles) and *Polarstern* (red star). Photographed by Svenja Kohnemann (d) Sea ice concentration on June 16, 2017 from NOAA/NSIDC (in percent), with the red dot showing the location of the ice floe station. A zoomed inset shows the drift track of the ice floe from 4 to 16 of June.

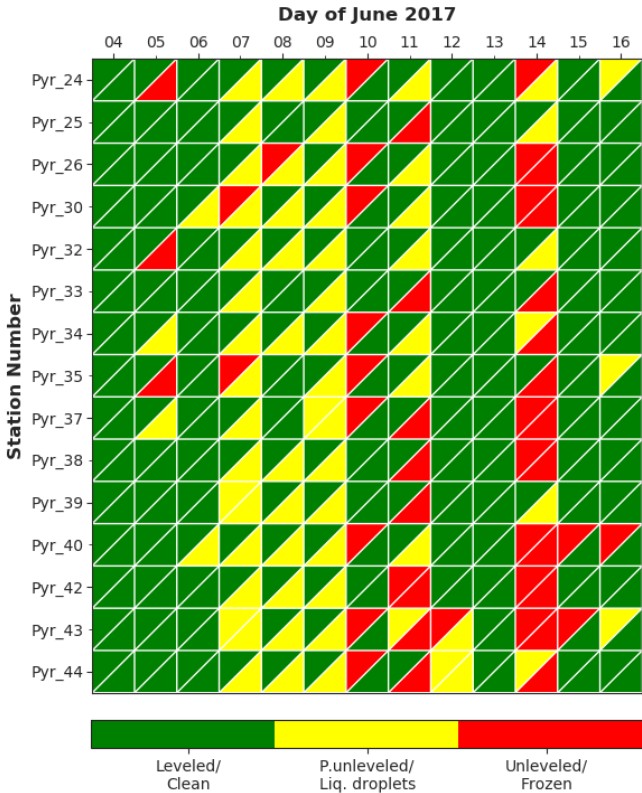

**Figure 2.** Quality flags for the pyranometer sensors: the pyranometer station number is shown on the y-axis, and date on the x-axis. Each square is divided into two triangles: the upper one shows the leveling flag, and the lower triangle the cleanliness status. Green is used for well-leveled and clean stations, while yellow denotes partially unleveled stations or the presence of liquid droplets on the domes, while red is used for unleveled stations or iced domes.

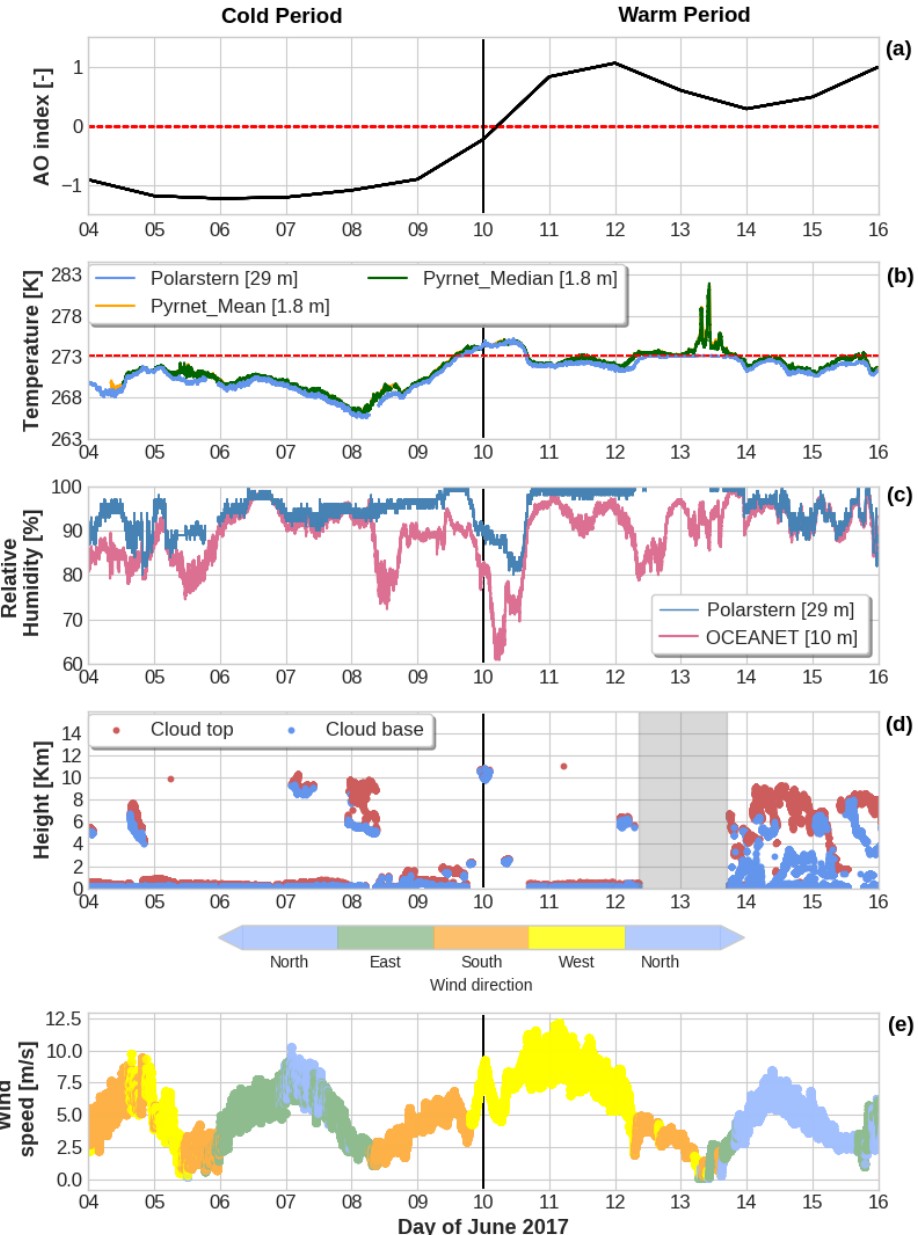

**Figure 3.** (a) AO index reported by the National Oceanic and Atmospheric Administration (NOAA). (b) Surface temperature from *Polarstern* meteorological instruments (blue), together with the mean (orange) and median (green) values from the pyranometer network. (c) Relative humidity (RH) from *Polarstern* (blue), and from the OCEANET container (pink). (d) Cloud base and cloud top height based on cloud radar and lidar. The gray background marks moments with no observations. (e) Wind speed (m/s) and direction obtained from *Polarstern* observations.


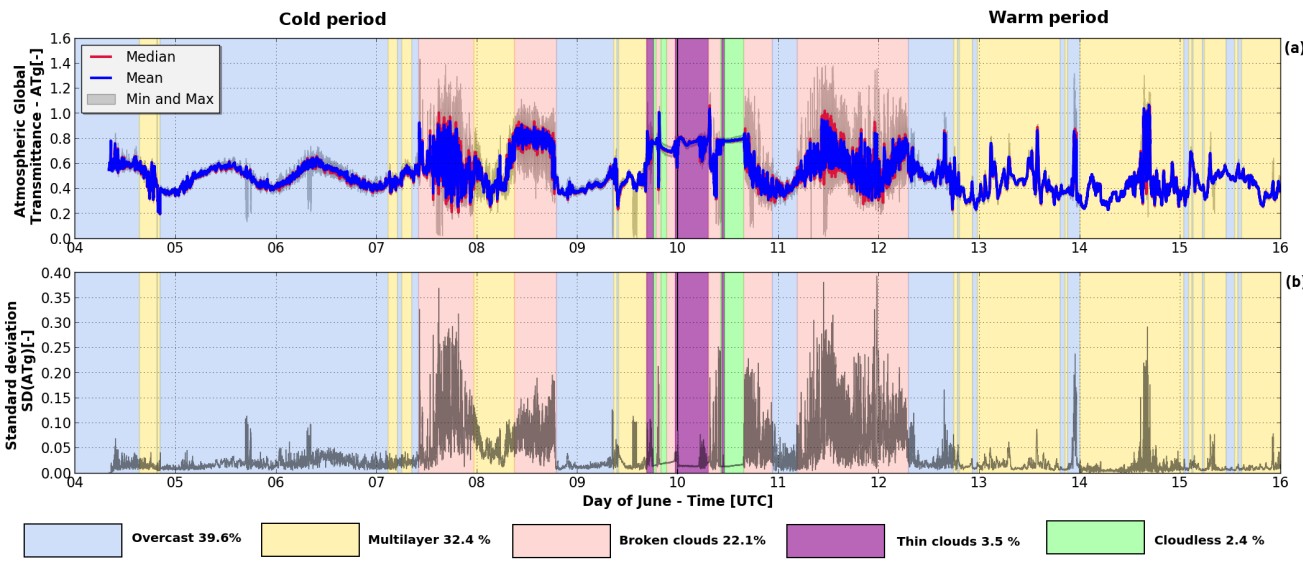

**Figure 4.** (a) Time series of atmospheric global transmittance (ATg) derived from pyraronmeter network. (b) Time series of the inter-station standard deviation of ATg.





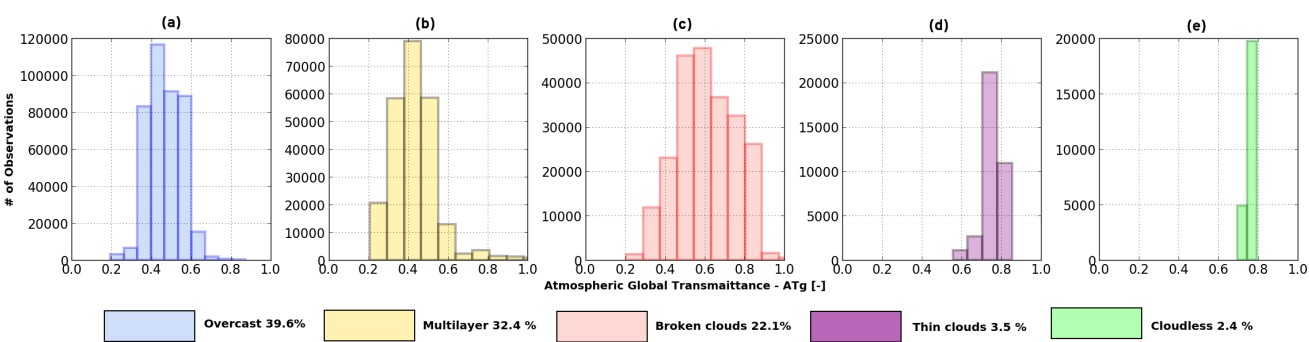

**Figure 5.** Histograms of atmospheric global transmittance (ATg) for (a) overcast, (b) multi-layer, (c) broken cloud, (d) thin cloud and (e) cloudless sky conditions.



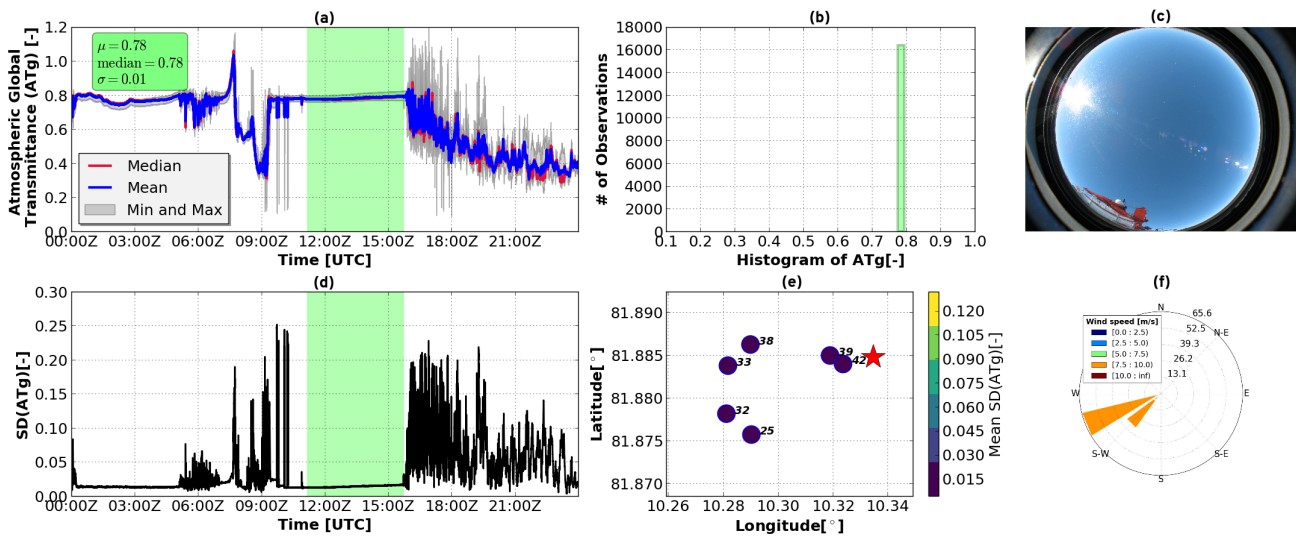

**Figure 6.** Overview of cloudless case: (a) shows the time series of atmospheric global transmittance (ATg), and the green-shaded background marks the cloudless period (June 10, 2017, 11:10Z - 15:43Z). (b) Histogram of global transmittance for cloudless conditions. (c) Photograph from the all-sky camera at 13:51:44Z. (d) Time series of the inter-station standard deviation (SD) of ATg based on all functional stations. (e) Map of the stations showing the temporal SD for individual stations, while the red star marks the position of *Polarstern*. (f) Wind rose for the selected period.





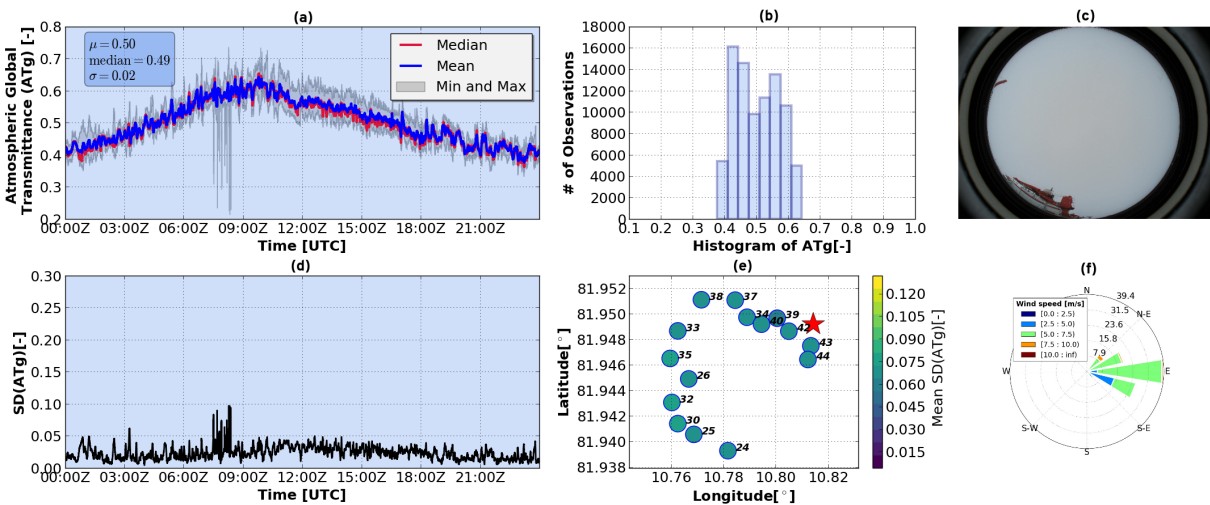

**Figure 7.** Overview of overcast case: Same as Fig. 6, but for June 6, 2017, 00:00Z - 23:59Z. All-sky camera photograph taken at 13:41:54Z.

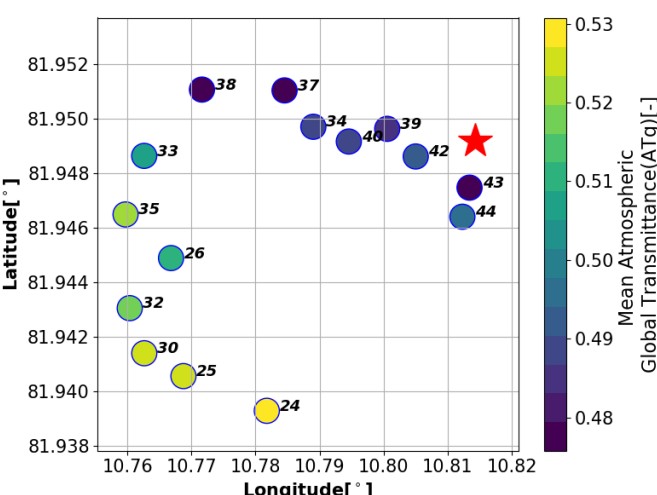

**Figure 8.** Station map showing the average atmospheric global transmittance for the overcast case (June 6, 2017, 00:00Z - 23:59Z). The red star marks the position of *Polarstern*.





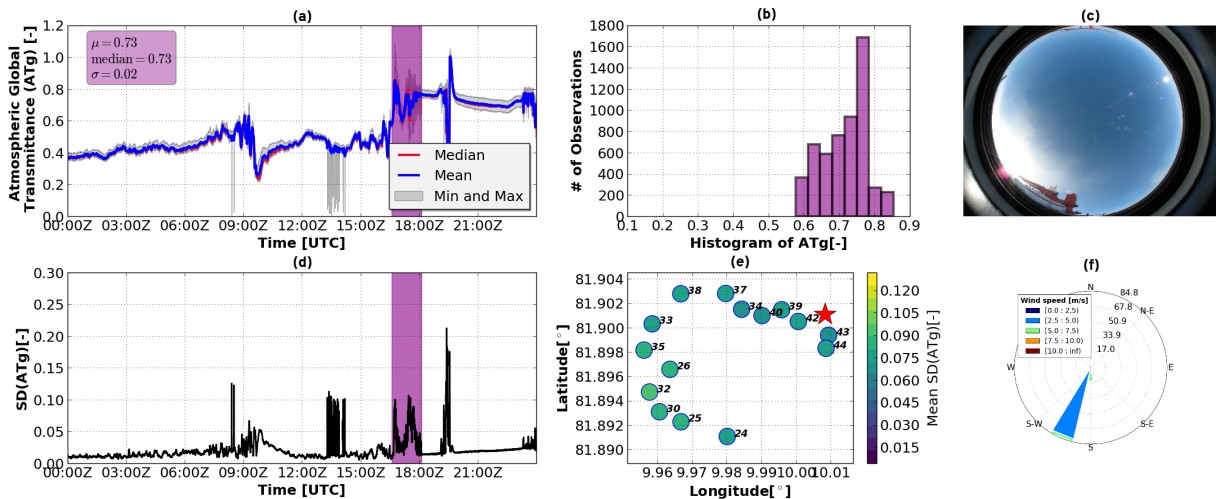

**Figure 9.** Overview of thin cloud case: Same as Fig. 6, but for June 9, 2017, 16:36Z - 18:08Z. All-sky camera photograph taken at 17:29:28Z.



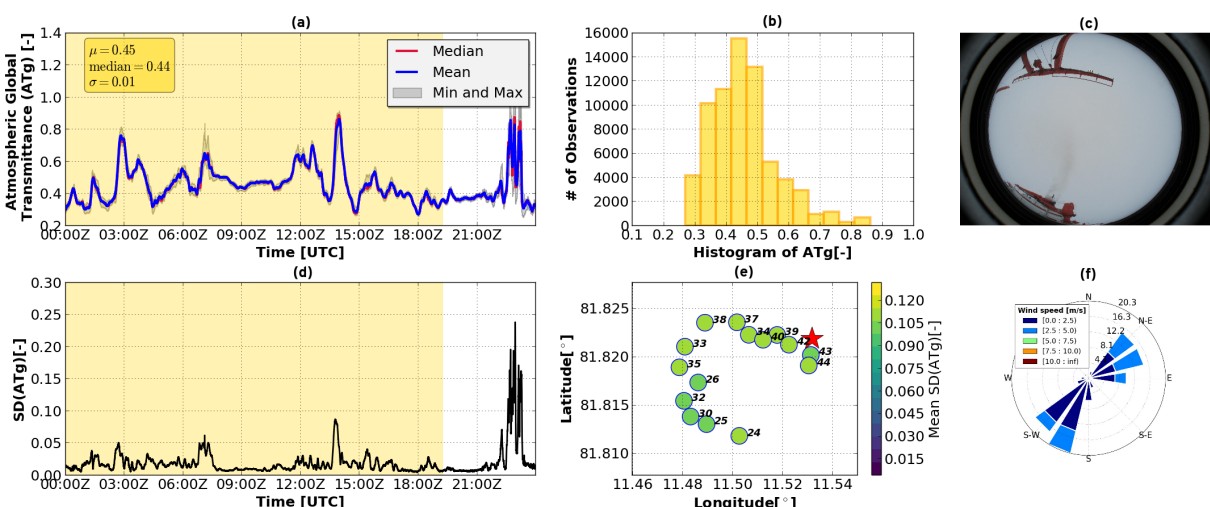

**Figure 10.** Overview of multi-layer case: Same as Fig. 6, but for June 13, 2017, 00:00Z - 19:15Z. All-sky camera photograph taken at 10:09:35Z.





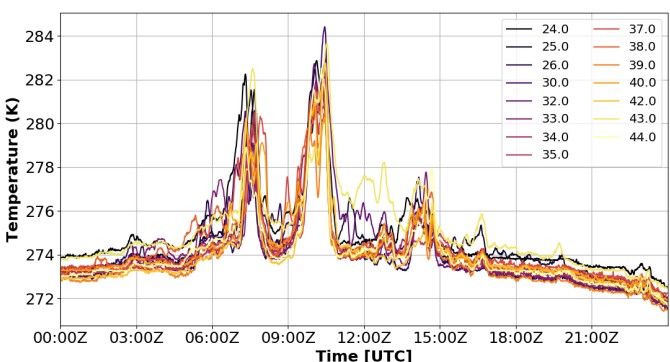

**Figure 11.** Time series of 1-minute averaged temperature for all stations on June 13, 2017.





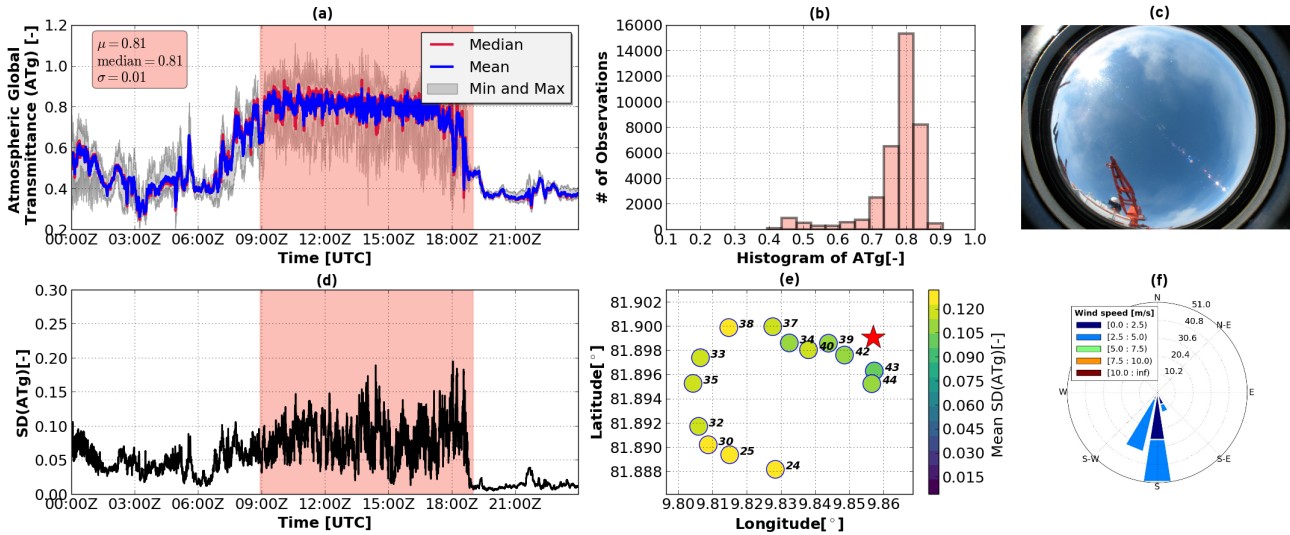

**Figure 12.** Overview of broken cloud case: Same as Fig. 6, but for June 8, 2017, 08:55Z - 19:00Z. All-sky camera photograph taken at 12:41:14Z.





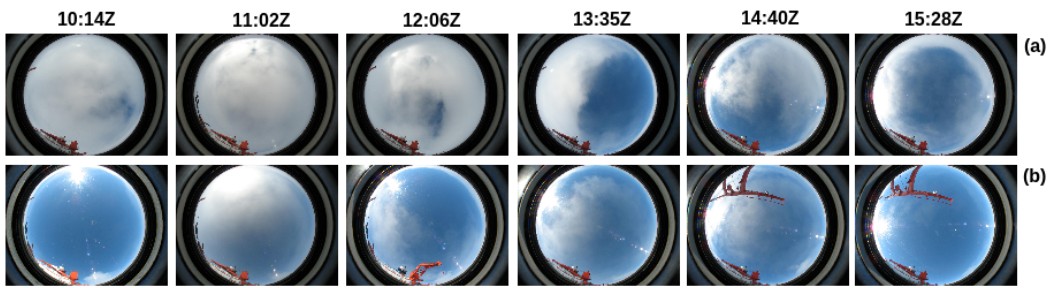

**Figure 13.** All-sky camera photographs for (a) June 7, and (b) June 8, taken at different times throughout the day.

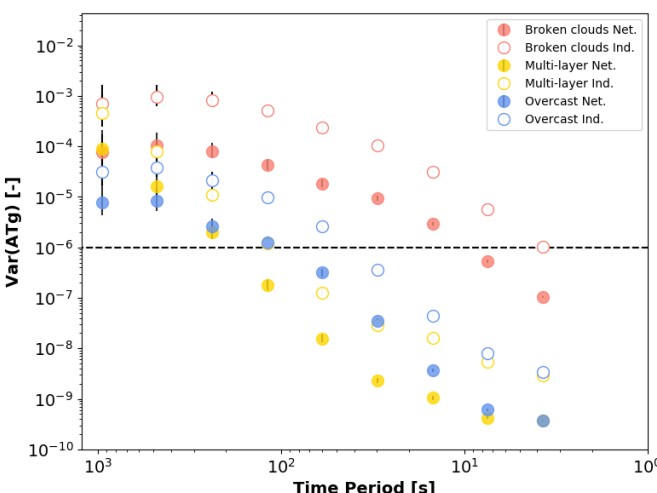

**Figure 14.** Wavelet-based power spectral density (WSD) of the station-averaged ATg (filled circles) and averaged WSD from the individual stations (empty circles), for 3-hour periods (15:00Z-18:00Z) with broken clouds (June 8, 2017), multi-layer (June 13, 2017) and overcast (June 6, 2017) conditions. The dashed horizontal line denotes the measurement uncertainty of the pyranometer.

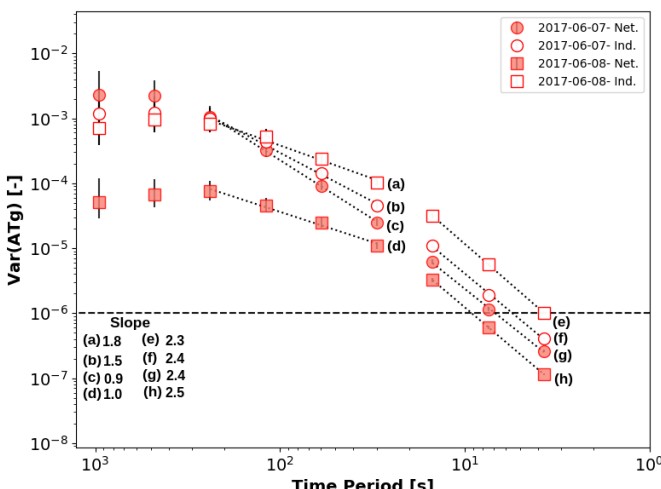

**Figure 15.** Wavelet-based power spectral density (WSD) of the station-averaged ATg (filled circles), and average WSD from the individual stations (empty circles) for 3-hour periods (15:00Z-18:00Z) of broken clouds on June 7 (circles), and June 8 (squares). Solid lines indicate the linear regression for the time period selected and the dashed horizontal line correspond to the measurement uncertainty of the pyranometer.