# Peer review of "Spatiotemporal variability of solar radiation introduced by clouds over the Arctic sea ice"

_Atmospheric Measurement Techniques, 2019_

## Referee Comment (RC1) · Anonymous Referee #1 · 29 Jul 2019

The comment was uploaded in the form of a supplement:
https://www.atmos-meas-tech-discuss.net/amt-2019-231/amt-2019-231-RC1-supplement.pdf
* * *

---

## Author Comment (AC1) · 15 Jan 2020

**Authors' response – AMT**

**Spatiotemporal variability of shortwave radiation introduced by clouds over the Arctic sea ice**
**by  Barrientos Velasco et al.**

Response to Anonymous Referee #1 (29 Jul 2019)

We would like to thank the Anonymous Referee # 1 for dedicating time and giving help to the improvement of the manuscript by providing us with valuable comments and suggestions.  We have now revised the initial submission, and hope that the manuscript is now acceptable for publication.

Our point-by-point response to the review comments is written here in bold font.

**Overall summary of major changes:**

We would like to inform the referee about the following major changes:

- Change of title for consistency
- Revision/restructuring of introduction due to suggestion of Referee #2
- Change of Figure 1 according to comments by Referee #2
- Improvement of the discussions section considering the comments by Referee #2
- Removal of table 1 due to suggestion of Referee #1

**Small clarifications**

- For consistency, we, now, refer to the shortwave component of the radiation as solar and to the longwave component as terrestrial radiation.
- A re-calculation was made of the area covered by the pyranometer network and the longitudinal extension of  value of 1.3 Km was re-adjusted to 1.59 Km.

**Specific comments:**
As mainly atmospheric global transmittance is discussed (and not global irradiance) the title could be changed.

**This suggestion has been discussed and we prefer to keep the title and change shortwave to solar just for consistency.**
**The reason to keep 'radiation' instead of 'transmittance' is additionally explain in the introduction with the following sentences:**

*'With the aim to better understand the spatial distribution of downward solar irradiance, we consider the solar atmospheric transmissivity as a proxy quantity to measure the influence of clouds on solar radiation, as it compensates at least to some degree for the influence of changes in solar elevation angle (Deneke et al., 2009)'*

Figure 2 indicates enormous problems with the horizontal leveling and/or with the cleanliness of the sensors on about half of the measurement days. What was the criteria to differentiate between an unleveled and a completely unleveled station?

**A more extensive description of the criteria used for the quality assurance is now given in section 2.1.2 and can be read as follows.**

*'The leveling criteria are based on the bubble position of the spirit level of the pyranometer. When the bubble was located inside the inner ring, the instrument has been considered as well-leveled, in between the two rings as partially leveled, and outside the ring as unleveled'*

Authors should further comment why days with liquid droplets on the domes were used in the analysis and how this probably influenced the results.

**Days with liquid droplet were considered because we wanted to include a larger amount of data that was not heavily compromised into the analysis. As we are mainly concerned with changes in transmissivity and not absolute values, we believe that these data are still useful for our analysis. It should be noted that the period of liquid droplets were likely to be relatively short, whereas frozen domes had the tendency to stay in that conditions for a longer period of time.**

**A corrected explanation is given in section 2.1.2. In the text the explanation is given as follows:**

*'The presence of liquid droplets is considered in the study due to their likely short residence time around the dome, and the fact that we have found observations to still be useful for our analysis. Furthermore, it is worth mentioning that during this likely short period, the presence of droplets is expected to cause a moderate underestimation of irradiance and more noisy observations.'*

As mentioned in the conclusions, the relevance of the results for the energy budget of the sea ice could not be assessed. The authors should describe in a little bit more detail in the outlook what would be necessary to do so.

We improved the explanation with the following text:

***'Future work will also be aimed at the investigation of radiative closure based on radiosonde soundings and ground-base remote sensing observations of cloud properties conducted aboard Polarstern as input to a 1-D radiative model for the entire PASCAL cruise. The output of this analysis will provide insights into the influence of clouds on the surface energy budget'***

Table 2 could by omitted.
**This table was omitted.**

In the following figures, a larger font should be used:
- Figure 1 (d)
- Figure 6 (f)
- Figure 7 (f)
- Figure 9 (f)
- Figure 10 (f)
- Figure 12 (f)

**The font was increased for all the figures above.**

**Technical corrections:**
- Page 2, Line 19: replace "sea-ice floe" by "sea ice floe".
- Page 3, Line 15: replace "Juelich" by Jülich".
- Page 4, Line 28: replace "better than than 2 %" by "better than 2 %".
- Page 6, Line 13: replace "Wendisch et al" by "Wendisch et al." This error occurs several times in the text.
- Page 11, Line 12: replace "dominated with by an anticyclonic" by "dominated by an anticyclonic".
- Page 12, Line 8: replace "Schade et al," by "Schade et al.,". This error occurs several times in the text.
- Page 19, Table 1: replace "responsivity per year)" by "responsivity per year"
- Page 21, Table 3: replace "Ambient temperature [Ta] , atmospheric global transmittance (ATg) [-]" by "Ambient temperature Ta [K], atmospheric global transmittance ATg [- ]"

**All the points above where changed and fixed.**

---

## Author Comment (AC2)

**Authors' response – AMT**

**Spatiotemporal variability of shortwave radiation introduced by clouds over the Arctic sea ice**
**by  Barrientos Velasco et al.**

Response to Anonymous Referee #2 (11 Nov 2019)

We would like to thank the Anonymous Referee # 1 for dedicating time and giving help to the improvement of the manuscript by providing us with valuable comments and suggestions.  We have now revised the initial submission, and hope that the manuscript is now acceptable for publication.

Our point-by-point response to the review comments is written here in bold font.

**Overall summary of major changes:**

We would like to inform the referee about the following major changes:

Change of title for consistency
Revision/restructuring of introduction due to suggestion of Referee #2
Change of Figure 1 according to comments by Referee #2
Improvement of the discussions section considering the comments by Referee #2
Removal of table 1 due to suggestion of Referee #1

**Small clarifications**

- For consistency, we, now, refer to the shortwave component of the radiation as solar and to the longwave component as terrestrial radiation.
- A re-calculation was made of the area covered by the pyranometer network and the longitudinal extension of  value of 1.3 Km was re-adjusted to 1.59 Km.

**Major comments:**

**1.** The introduction section feels somewhat disjointed to the reader.
A rather broad and unfocused overview of Arctic sea ice changes is followed by a brief mention of the PASCAL campaign, followed by a longer description of projects and measurement campaigns with the same sensors in Germany (which seems quite unconnected to the topic at hand), followed by a relatively suddenly appearing statement that the goal of the paper is the "analysis of the temporal and spatial variability of the atmospheric global transmittance (ATg)". What is the main point that the authors want to make here? Why is the analysis of ATg relevant, what does it actually relate to, what is the long-term goal behind the work? I recommend a review and revision of the introduction to make it more focused on the task at hand and its scientific justification/background.
**The introduction was revised and hopefully improved to address this point. The main focus of the study is presented in the second, fourth and last paragraph of Introduction, and some connecting sentences were added in between. PASCAL has not been emphasized more, because there is already an overview paper and a complete expedition report published which are cited in our text.**

**2.** From pg. 4, ln 12, it appears that all the data presented correspond to the waveband 0.3 – 1.1 microns, correct? So the calculated ATg is not the full shortwave broadband transmittance, but rather the visible-NIR section of it? Which would imply that the wavelength-dependent effects (e.g. Nann & Riordan, 1991 for some discussion) of clouds on the SWIR part of the solar irradiance waveband are not measured and their impacts on ATg variability remains unknown? Why does the discussion section contain no content on this point?
**To answer this aspect, we have now stressed more the limitations of the spectral range (0.3-1.1 microns) of the pyranometer network in the instrumental section and further explain how the re-calibration using a broadband pyranometer tries to compensate plausible discrepancies between the pyranometer used in our experiment and a broadband pyranometer. We acknowledge the existence of some deviations, however for the analysis of the variability this aspect likely does not matter.**

**The text added in section 2.1 of the manuscript is the following:**

*'The spectral range of the pyranometer network neglects the spectral irradiance beyond 1.1 μm which comprises about 22% of the incoming solar energy (Nann and Riordan, 1991), however based on previous studies it was demonstrated that the spectral range where cloud transmittance is higher occurs between 0.3-0-7 μm (Wiscombe et al. 1984, Barlett et al., 1998). Therefore, our set up is still expected to capture the main variability effects of cloud transmittance.'*

**The text added in section 2.1.1 of the manuscript is the following:**

*'In order to update the calibration of the sensors and account for the spectral difference between a broadband pyranometer and the pyranometer network, inter-comparison measurements were conducted in May 2018.'*

To further address this point we also included an additional paragraph in the discussion section. The paragraph added can be read below.

*'As one specific aspect of the solar radiation budget, the present study focuses on the analysis of the spatiotemporal variability of the solar atmospheric transmissivity at the surface as it is introduced by clouds. Despite the fact the silicon photodiode pyranometers used in this study operate with a limited spectral range of 0.3-1.1 μm and thus do not cover the entire solar spectral range like a conventional broadband thermopile pyranometer, they do capture the main changes of the solar spectral transmission induced by clouds (Barlett et al., 1998). Therefore, it is worth stressing that the analysis of the spatiotemporal variability induced by clouds in the shortwave infrared region (e.g. in the atmospheric windows at 1.6 or 2.2 μm commonly used for satellite remote sensing) is outside of the scope of this study, and might be a valuable investigation for future research.'*

**3.** The middle part of section 3.1. is confusing when comparing with Figure 3. First, Figure 3 and the later text on this page defines the warm period as June 4-9. The text on pg 7, lines 11-12 is different, why? Second, Figure 3 appears to define the warm period as a period of positive AO, in conflict with the text and present definitions of AO. Figure 3a needs to be checked and revised.
**The interpretation of AO and connection with near surface temperature has been revised. The classifying the period as warm does not mean that it is warmer than usual in an inter-annual mean context. It means that during the period of study, the temperature was warmer than the local average for the considered period, however this local warm conditions are not warmer than the inter-annual mean temperature for the Arctic. Also few aspects of the text were changed to avoid any confusion.**

**- The section 3.1 subtitle was changed to** *'Near-surface temperature classification during PASCAL: Ice floe camp and synoptic implications'*

**- The classification is merely made considering the near-surface air temperature, and the AO index is introduced to describe the inter-annual and synoptic components to the analysis. Therefore we include the following text:**

*'The classification made is solely based on the near-surface air temperature and the AO index reveals the synoptic patterns considering the interannual variability'*

**4.** The explanation in section 3.7. on the ATg variability causes for broken clouds do not feel convincing. It could easily be argued that the areas further away from floe edge have a thicker and dryer snow cover with higher albedo, thus making it possible that enhanced multiple cloud-surface scattering could contribute significantly to observed variability in addition to cloud organization? The case needs to be made better on this point.
**We agree with the reviewer that it is not possible to attribute all the variability to the broken cloud conditions, since part of it is also might result from cloud-surface reflections. Therefore, we rephrased part of the text to do not mislead the reader.**

*'The increase of diffuse solar radiation is attributed not only to the broken cloud conditions, but also to the multiple reflection between surface and heterogeneous cloud fields'*

**5.** Section 3.8. is a bit underwhelming content-wise. The finding that irradiance variability decreases with shorter observation time periods is quite self-evident, as is the fact that broken cloud cases exhibit the largest variability, and that thick multilayer clouds show the least variance. This reviewer would at least challenge the authors to take their thinking a step further. What do these results mean for recommendations on how to measure irradiance on snow and ice in the future? How much more "wrong" would you be if you had just a single pyranometer doing the work of the network?

**To better address the points mentioned we included two paragraphs in section 3.8 and supplentary explanation in the dicussion and conclusion sections.**

**The paragraphs added to section 3.8 can be seen bellow:**

*'Comparing Figure 14 and Figure 15, differences of the WSD for the average of all functional stations (filled circles) and considering the individual point-like measurements (empty circles) can be seen, indicating that the variability is reduced by about 0.1 as the spatial averaging cancels out part of the small-scale spatial variability. Hence, spatial averaging of stations allows us to better resolve temporal changes, while the magnitude of the difference provides information on the small-scale spatial structure of clouds. The results obtained here also suggest that future measurements of GHI in similar highly reflective surface conditions should be done with a temporal resolution of at least 10 seconds to fully capture the variability under broken cloud conditions.'*

**We also added additional comments related to this point in the *'Discussion, conclusion and outlook'* section. The text can be read bellow:**

*'Ideally, it is recommended to have complementary observations of the surface albedo at the same spatial and temporal resolution as the pyranometer network. In this way, it would be possible to better quantify the spatial variability induced by the multiple reflections between the highly reflective surface and clouds, and the effects of inhomogeneities in surface albedo. This would also help to better understand surface features visible in Figure 8, which can only be observed with setups like the pyranometer network used here. Additionally, in the future, experiments including one or several spectrometers can provide information to quantify changes in the spectral distribution of solar radiation under different sky and surface conditions. With observations extending into the shortwave infrared range, a similar analysis could help to better understand cloud effects on atmospheric transmissivity at wavelengths above 1.1 μm.*

*'The pyranometer network offers valuable information on the variability of the solar irradiance at the surface on small scales, making it possible to better characterize temporal and spatial fluctuations than by single station measurements. In the future, we plan to use this dataset to better understand modulation of the downwelling solar irradiance considering the effects of the horizontal distribution of clouds, the solar zenith angle, cloud phase, surface reflectivity. With information on wind direction and speed, it might be possible to separate the observed variability into components*

*arising from advection, by cloud spatial structure, and by temporal changes of clouds. This dataset shall be used as a reference for a comparison with radiative transfer simulations using a 3-D Monte Carlo radiative transfer model using large eddy simulations as input, which are also being conducted within the scope of the (AC)³ project. This will allow us to further investigate the link between cloud spatial structure and the resulting variability in the solar radiation field. Future work will also be aimed at the investigation of radiative closure based on radiosonde soundings and ground-base remote sensing observations of cloud properties conducted aboard Polarstern as input to a 1-D radiative model for the entire PASCAL cruise. The output of this analysis will provide insights into the influence of clouds on the surface energy budget.'*

**6.** Figure 1 has too much content squeezed into a single frame. At the very least, replace Fig 1d with a zoomed in region around Svalbard, with the drift track marked clearly. The sea ice concentration data has little to do with the manuscript. Also considering flipping the photograph so that the relative directions match with subplot b –the best quality would result from combining the two if the photograph has enough auxiliary information to make it a geotiff **Figure 1 was changed. The sea ice concentration is not shown anymore. We flipped the plot, but the photograph is the same as before. (See bellow)**

[Figure]

**Figure 1.** (a) Photograph of a pyranometer station on the ice floe. (b) Map of the pyranometer stations. Red circles show the location on June 11, 2017, at 14:50Z, while the red star marks the position of *Polarstern*, and the turquoise hexagon marks the approximate position of a melt-pond. Note that the latitude and longitude axes have been inverted for easier comparison with panel (c). (c) Edited photograph of the ice floe station showing the approximate location of several stations (red circles) and *Polarstern* (red star). Photographed by Svenja Kohnemann (d) Drifting track of the ice floe from 4 to 16 of June.

**Minor comments (page – line):**
**2 – 2:** extended -> extensive. **Changed**
**2-2:** "As the surface temperature increases, snow and sea ice melts,..." -> careful here with the wording. While brief warm periods can occur over the Arctic sea ice pack, most melt occurs at close to 0 C because the melt processes eat up the excess energy. Recommend simplifying to "Initialization of snow and ice melt reduces the surface albedo and increases the amount of...".
**Comment considered and text changed to:**

*'When the surface temperature reaches 273.15 K, snow and sea ice start to melt, reducing the albedo and increasing the amount of shortwave radiation absorbed by the surface, a process known as the ice-albedo feedback (Curry et al., 1995)'*

**4-15:** "Larger than the accuracy achieved..." -> a larger accuracy means a poorer measurement quality? Be precise with the terms, please. Consider revising the ambiguous "accuracy" (consists of both bias and precision) with "measurement uncertainty". **Changed, see text bellow.**

*'It is worth mentioning that due to the large number of stations, a low-cost pyranometer was used, with an expected accuracy of about 5 %, which is substantially larger than measurement uncertainty achieved by state-of-the art secondary standard thermopile pyranometers.'*

**6-1:** ATg presented are the averages from the pyranometer network? Define this in the text, please.
**It does not imply a mean value. It shows the methodology to derive ATg. Mean values are shown and mentioned in the data analysis (section 3).**

**6-9:** Lateral cloud edge reflection increases irradiance in broken cloud cases, but that is primarily a single-scattering mechanism. It's not only multiple scattering events, so be precise, please.

**Comment considered and text changed to:**

*'Under certain situations however, the presence of broken clouds can amplify ATg to reach values larger than 1 due to horizontal photon transport and 3D radiative effects'*

**Figure 6d:** The variability in ATg mentioned on pg 10, ln 1, is invisible without altering the color mapping

**We want to present all plots with the same color scale for easier comparison, and thus chose this scaling intentionally. The difference is not visible because it is indeed very small.**

**11 – 27:** The spatial variation of ATg may be low; ATg itself appears to vary a lot from 0.2 to 0.6. Revise for precision.

**The variation is due to the variation of the structure of the multilayer clouds. The thicker the double layer cloud, the lower the resulting ATg. This has been rephrased in the paper for clarity.**

*'The variation observed is mainly attributed to the different vertical structure of the multiple cloud layers (See. Fig. 10b, Table 2)'*